# Integrated Physiological, Transcriptomic, and Metabolomic Analyses Revealed Molecular Mechanism for Salt Resistance in Soybean Roots

**DOI:** 10.3390/ijms222312848

**Published:** 2021-11-27

**Authors:** Jie Jin, Jianfeng Wang, Keke Li, Shengwang Wang, Juan Qin, Guohong Zhang, Xiaofan Na, Xiaomin Wang, Yurong Bi

**Affiliations:** 1Ministry of Education Key Laboratory of Cell Activities and Stress Adaptations, School of Life Sciences, Lanzhou University, Lanzhou 730000, China; jinj2015@lzu.edu.cn (J.J.); likk17@lzu.edu.cn (K.L.); wangshw16@lzu.edu.cn (S.W.); qinj18@lzu.edu.cn (J.Q.); naxf@lzu.edu.cn (X.N.); 2State Key Laboratory of Plateau Ecology and Agriculture, Qinghai University, Xining 810016, China; wangjf12@lzu.edu.cn; 3Center for Grassland Microbiome, Collaborative Innovation Center for Western Ecological Safety, State Key Laboratory of Grassland Agro-Ecosystems, College of Pastoral Agriculture Science and Technology, Lanzhou University, Lanzhou 730000, China; 4Institute of Dryland Agriculture, Gansu Academy of Agricultural Sciences, Lanzhou 730070, China; zgh@gsagr.ac.cn

**Keywords:** transcriptome, metabolome, amino acid, nitrogen metabolism, nitrogen use efficiency, TCA cycle, salt stress, soybean

## Abstract

Salinity stress is a threat to yield in many crops, including soybean (*Glycine max* L.). In this study, three soybean cultivars (JD19, LH3, and LD2) with different salt resistance were used to analyze salt tolerance mechanisms using physiology, transcriptomic, metabolomic, and bioinformatic methods. Physiological studies showed that salt-tolerant cultivars JD19 and LH3 had less root growth inhibition, higher antioxidant enzyme activities, lower ROS accumulation, and lower Na^+^ and Cl^-^ contents than salt-susceptible cultivar LD2 under 100 mM NaCl treatment. Comparative transcriptome analysis showed that compared with LD2, salt stress increased the expression of antioxidant metabolism, stress response metabolism, glycine, serine and threonine metabolism, auxin response protein, transcription, and translation-related genes in JD19 and LH3. The comparison of metabolite profiles indicated that amino acid metabolism and the TCA cycle were important metabolic pathways of soybean in response to salt stress. In the further validation analysis of the above two pathways, it was found that compared with LD2, JD19, and LH3 had higher nitrogen absorption and assimilation rate, more amino acid accumulation, and faster TCA cycle activity under salt stress, which helped them better adapt to salt stress. Taken together, this study provides valuable information for better understanding the molecular mechanism underlying salt tolerance of soybean and also proposes new ideas and methods for cultivating stress-tolerant soybean.

## 1. Introduction

Salt stress is a major restricting factor in agricultural production [1,2]. The salinized soil currently accounts for 8% of the world’s total land area, and it is expected that the area of irrigated agriculture and salt-affected farmland in semi-arid areas will double by 2050 [3]. In order to meet the fast-growing food demand for the global population, it is estimated that food production needs to increase by 70–110% to maintain current levels by 2050 [4]. Therefore, it is particularly crucial to enhance the salt tolerance of conventional crops without expanding agricultural land area [5].

Soybean (*Glycine max* L.), as one of the widely planted crops, accounts for about 56% of the total oilseed production in the world [6,7]. However, soybean is especially sensitive to salinity, and its yield is decreased by nearly 40% with the increased salt stress [8,9]. On the one hand, salinity stress can reduce the survival rate of rhizobia in the soil, thus inhibiting symbiotic nitrogen fixation and resulting in the reduction in soybean yield. On the other hand, the host plants are more salt-sensitive than rhizobia, which is a main limiting factor of yield under salt stress [10]. Therefore, improving the salt tolerance of soybean seedlings is vitally important to improve soybean production.

As the most soluble salt in soil, NaCl causes damages to plants in many diverse ways, including osmotic stress, ionic stress, oxidative stress, and nutritional disorders [11,12,13]. To cope with adverse high salinity environment, plants have to employ several adaptive strategies, for example, by inducing enzymatic scavengers, such as superoxide dismutase (SOD), peroxidase (POD), catalase (CAT), and ascorbate peroxidase (APX), to mitigate ROS stress and by decreasing Na^+^ content and increasing K^+^ content in the cytoplasm to keep the high K^+^/Na^+^ level [14,15,16,17,18]. In addition, high salinity also alters many metabolic processes. The accumulation of proline, betaine, and sucrose is involved in the osmotic regulation of plants under salt stress [19]. Many organic acids, fatty acids, and phytohormones are changed to adapt to saline conditions [20]. It has been reported that the exogenous application of abscisic acid could ameliorate wheat growth under salinity stress [21]. Moreover, accumulation of free amino acids was observed in Arabidopsis exposed to salt stress [22]. Zhang et al. [23] demonstrated that seven amino acids were significantly enriched in tomatoes during the adaptation to salt stress.

Currently, the plant responses to salt stress have been widely studied at molecular and physiological levels. More and more molecular mechanisms have been revealed along with the progress of genomics, proteomics, and metabolomics. Huang et al. [24] used multi-omics analysis to reveal the mechanisms of salt resistance in sea barley. Xiong et al. [25] conducted transcriptomic analysis in alfalfa to clarify its salt adaptive strategies by up-regulating hormone signal transduction and the antioxidant gene expression. Zhang et al. [26] explored the different responses to salinity in sesame through metabonomic analysis and revealed that amino acid metabolism, sucrose, raffinose, and oligosaccharide metabolism were enhanced in salt-tolerant genotype. In this study, three soybean cultivars (JD19, LH3, and LD2) with different salt tolerance were used to investigate the mechanism and specific salt adaptive strategies. We integrated physiology, transcriptomic, and metabolomic analysis of JD19, LH3, and LD2 exposed to salinity stress to identify the key genes, metabolites, and metabolic pathways potentially contributing to salt tolerance and further verify the crucial metabolic pathways. The present study will provide valuable information for improving salt resistance in soybean and lay a foundation for crop improvement in the future.

## 2. Results

### 2.1. Differences of Three Soybean Cultivars Response to Salt Stress

It is well known that salt stress can change root architecture and biomass allocation [18]. As shown in Figure 1A, the root growth was different under salt stress among the three soybean cultivars. The primary root length in JD19 and LH3 was greater than that of LD2. A total of 100 mM NaCl treatment reduced the primary root length of JD19, LH3, and LD2 by approximately 19.8%, 20.7%, and 36.8%, respectively (Figure 1B). The numbers of lateral roots did not exhibit significant differences among LH3, LD2, and JD19 under the control condition. However, 100 mM NaCl treatment significantly decreased the lateral root numbers in all three cultivars, especially in LD2 (Figure 1C). Although the fresh weight and the dry weight in all cultivars were decreased under salt stress, JD19 and LH3 obviously lost less biomass than LD2 (Figure 1D,E). These results suggested that JD19 and LH3 were more tolerant to salinity than LD2.

To investigate whether ROS level is associated with different tolerance of three soybean genotypes, the H_2_O_2_, MDA contents, and antioxidant enzyme activities were analyzed. The activities of SOD, CAT, POD, and APX did not show significant differences among the three soybean cultivars without NaCl treatment. However, salt stress strongly increased the activities of the four antioxidant enzymes in roots, especially in JD19 and LH3 (Appendix A). On the contrary, the contents of H_2_O_2_ and MDA were higher in LD2 than in JD19 and LH3 under salt stress (Appendix A). Thus, high activities of antioxidant enzymes and low contents of H_2_O_2_ and MDA might be one of the reasons for salt tolerance in JD19 and LH3. Moreover, salt stress always leads to ion toxicity for plants. Figure 2 showed that ion contents did not show a significant difference among the three soybean cultivars without NaCl treatment. However, salt stress strongly stimulated Na^+^ and Cl^-^ accumulation in three soybean roots and reduced the K^+^ content (Figure 2). In comparison with JD19 and LH3, LD2 accumulated more Na^+^, Cl^−^ and less K^+^ contents in the root, which finally resulted in a higher level of Na^+^/K^+^ ratio under salt stress. The Na^+^/K^+^ ratio in JD19, LH3, and LD2 increased by 24.1%, 24.5%, and 32.9%, respectively (Figure 2C).

### 2.2. Transcriptional Characteristics of Three Soybean Cultivars Response to Salt Stress

To gain further insight into the salt tolerance mechanism of soybean, root tissues of three soybean cultivars were collected with control and 100 mM NaCl treatment for transcription analysis. The mapping of clear reads to the soybean genome showed that 89.41%–93.76% of reads aligned to only one location in the reference genome (Appendix A). A threshold of fold change ≥ 2 and FDR ≤ 0.05 was used to identify differentially expressed genes (DEGs). As shown in Figure 3A, a total of 1482 DEGs were identified in JD19, of which 837 genes were up-regulated, and 645 genes were down-regulated. A total of 1368 DEGs were identified in LH3, of which 739 genes were up-regulated, and 629 genes were down-regulated, and a total of 2116 DEGs were identified in LD2, of which 833 genes were up-regulated, and 1283 genes were down-regulated. Additionally, after 100 mM NaCl treatment, 123 genes were induced, and 117 genes were repressed in all soybean genotypes (Figure 3B,C). A total of 183 up-regulated genes and 101 down-regulated genes were overlapped between the salt-tolerant genotypes JD19 and LH3 (Figure 3B,C). To test the RNA-seq data, we randomly selected five genes for qRT-PCR validation. The result of these genes by qRT-PCR was highly correlated with the RNA-seq data, which further proved the reliability of transcriptomic results (Appendix A).

### 2.3. Differentially Expressed Genes among JD19, LH3, and LD2

Hierarchical clustering analysis showed that the commonly regulated genes exhibited differential expression among JD19, LH3, and LD2 in the saline condition (Appendix A). GO and KEGG pathway enrichment analysis was used to explore the underlying function of key salt-responsive genes in soybean. The biological process of GO terms showed that the 123 up-regulated core genes in all cultivars were enriched in ammonium transmembrane transport, nitrogen utilization, sulfate assimilation, oxidation-reduction process, cellular redox homeostasis, response to oxidative stress, flavonoid biosynthesis, flavonoid metabolism, and small molecule metabolism. Biological processes of carbohydrate transmembrane transport, organic substance transport, transmembrane transport, adenine salvage, nitrogen compound transport, carbohydrate metabolic process, monovalent inorganic cation transport, citrate transport, anion transport, and regulation of transcription, DNA templated were enriched in 117 down-regulated core genes (Appendix A). KEGG pathway enrichment analysis revealed that genes involved in protein processing in glyoxylate and dicarboxylate metabolism, zeatin biosynthesis, carbon metabolism were up-regulated by salt stress (Appendix A). The together down-regulated DEGs in the three cultivars did not have the same significant enrichment of the KEGG pathway.

JD19 and LH3, as two salt-tolerant genotypes, jointly regulated some DEGs under salt stress, suggesting that these genes may function in salt tolerance of JD19 and LH3. GO term enrichment analysis revealed that the categories of glutathione metabolism process, response to stress, organic cation transport, transmembrane transport, and peptide metabolic process were enriched among genes that were up-regulated in JD19 and LH3 under salt stress (Figure 3D). Meanwhile, the categories of nodulation, regulation of RNA biosynthetic process, regulation of RNA metabolic process, glycerophospholipid metabolic process, purine-containing compound metabolic process were enriched among down-regulated genes in JD19 and LH3 under salt stress (Figure 3E). KEGG enrichment analysis showed that the pathways of glycine, serine, and threonine metabolism, glutathione metabolism, pentose phosphate pathway, tryptophan metabolism were enriched in up-regulated DEGs in JD19 and LH3 (Figure 3F). Meanwhile, the pathways of caffeine metabolism, fructose and mannose metabolism, and plant hormone signal transduction were enriched in down-regulated DEGs (Figure 3G).

In addition to the jointly regulated DEGs, we also found that JD19, LH3, and LD2 have some independently regulated DEGs under salt stress, which may be one of the reasons for the difference in salt tolerance among the three cultivars. During salt stress, a total of 411 genes were independently up-regulated, and 254 genes were down-regulated in JD19. Likewise in LH3 and LD2, there are 377 and 534 up-regulated genes and 297 and 879 down-regulated genes, respectively (Figure 3B,C). GO analysis revealed that JD19 independently up-regulated DEGs were enriched in biological processes related to oxalate metabolic process, carbon fixation, ribonucleoprotein complex assembly, organic anion transport. LH3 independently up-regulated DEGs were enriched in biological processes related to glycerol ether metabolic process, glycolipid transport, regulation of secretion, regulation of exocytosis. LD2 independently up-regulated DEGs were enriched in biological processes related to L-phenylalanine catabolic process, unsaturated fatty acid biosynthetic process, nitrogen fixation, mitochondrial respiratory chain complex assembly (Appendix A). Moreover, JD19 independently down-regulated DEGs were enriched in biological processes related to cellular component biogenesis, sulfur compound catabolic process, glycolipid metabolic process. LH3 independently down-regulated DEGs were enriched in biological processes related to auxin efflux, auxin homeostasis, auxin polar transport. LD2 independently down-regulated DEGs were enriched in biological processes related to the regulation of jasmonic acid-mediated signaling pathway, hydrogen peroxide catabolic process, lipid homeostasis (Appendix A). KEGG pathway enrichment analysis showed that JD19 independently up-regulated DEGs were enriched in a pathway related to lysine degradation, ABC transporters, circadian rhythm-plant, peroxisome, arginine, and proline metabolism. LH3 independently up-regulated DEGs were enriched in a pathway related to isoflavonoid biosynthesis, alanine, aspartate, and glutamate metabolism. LD2 independently up-regulated DEGs were enriched in a pathway related to taurine and typotaurine metabolism, pyruvate metabolism, and starch and sucrose metabolism (Appendix A). Meanwhile, JD19 independently down-regulated DEGs were enriched in pathways related to base excision repair, tropine, piperidine and pyridine alkaloid biosynthesis, and isoquinoline alkaloid biosynthesis. LH3 independently down-regulated DEGs were enriched in pathways related to β-alanine metabolism, nitrogen metabolism, and regulation of autophagy. LD2 independently down-regulated DEGs were enriched in pathways related to arginine and proline metabolism, linoleic acid metabolism, valine, leucine, and isoleucine degradation (Appendix A). The above results indicate that the salt-tolerant cultivars have both common pathways and unique pathways to deal with salt stress.

### 2.4. Different Salt-Responsive Genes among JD19, LH3, and LD2

Among the up-regulated DEGs, 18, 20, and 10 genes were annotated into glutathione metabolism and stress response-related metabolic processes in JD19, LH3, and LD2, which were mostly associated with response to salt stress (Figure 4A). Three DEGs were co-expressed in all cultivars, encoding glutathione S-transferase (GSTU37), small molecule heat shock protein (LOC100170732), and ascorbic acid peroxidase (LOC100792735). The expression of LOC100170732 and LOC100792735 were significantly increased in three soybean cultivars, while the expression of GSTU37 was decreased in LD2 but increased in JD19 and LH3 under salt stress (Figure 4A). Interestingly, we found that four genes, namely glutathione reductase (LOC100783766), CAT5, and late embryogenesis abundant proteins (LEA2 and LEA5), were significantly induced by salt stress in JD19 and LH3. In addition, five and seven genes encoding glutathione S-transferase were specifically expressed in JD19 and LH3, respectively (Figure 4A).

Under the salt stress condition, 15, 17, and 14 genes were annotated into multiple amino acid metabolism pathways in JD19, LH3, and LD2, respectively (Figure 4B). A total of 100 mM NaCl treatment inhibited ornithine decarboxylase (LOC548088) and proline dehydrogenase (PDH) genes, while asparagine synthase (AS2), serine hydroxymethyltransferase (LOC100499634), and glycine lyase (LOC100785811) genes were activated (Figure 4B). The hydroxymethyltransferase (SGT3), phosphoserine aminotransferase (LOC100795286), and glycine lyase (LOC100306654) genes were significantly up-regulated only in JD19 and LH3. In addition, most of these DEGs were independently regulated in three cultivars under salt stress. For example, two genes, LOC100797464 and LOC100800978, encoding proline dehydrogenase were significantly down-regulated in JD19, while LOC100797583 and LOC100780526 encoding alanine glyoxylate aminotransferase were significantly up-regulated in LH3. The expression of glutamate synthase (LOC100789509, LOC100801991, LOC100812201, GDH1) and low specific L-threonine aldolase (LOC100800811, LOC100790890) were significantly inhibited by salt stress in LD2 (Figure 4B).

### 2.5. Metabolic Characteristics of Soybean Response to Salt Stress

In order to explore alterations in major metabolic pathways in response to salt stress in different cultivars, we performed a metabolomic analysis. Principal component analysis (PCA) showed that the PC1 explained 15% of the total variation, while the PC2 and PC3 explained 11% and 9.7% variation across the metabolites data, respectively (Figure 5A). The results indicated that there were obvious differences between the different treatment groups of the three soybean cultivars. Orthogonal projections to latent structures-discriminant (OPLS-DA) analysis was performed by screening out orthogonal variables unrelated to the classification variables. Through the analysis of non-orthogonal and orthogonal variables, reliable metabolic differences were obtained. The results showed that JD19, LH3, and LD2 were clearly separated under salt treatment, with all groups falling in the 95% confidence intervals (Hotelling’s T-squared ellipse) (Figure 5B–D). Therefore, it was speculated that metabolic characteristics were significantly different among the three cultivars under salt stress. In the OPLS-DA model, Q^2^ (cum) was used to estimate the ability of prediction, and R^2^Y (cum) was used to estimate the “goodness of fit” of the model [27]. The results indicated that the test model was credible and repeatable.

### 2.6. Differences in Metabolite Content among JD19, LH3, and LD2

The identified differentially expressed metabolites (DEMs) were screened by variable importance in the VIP (VIP > 1) score and t-test (*p* < 0.05) based on the OPLS-DA model. In total, there were 14 DEMs in the three soybean cultivars under salt stress (Appendix A). The heat map results indicated that all DEMs had obvious clustering, and the content of most compounds changed clearly under salinity stress in these cultivars (Figure 6A–C). Interestingly, the contents of oxproline, 4-aminobutyric acid, malonic acid, and beta-alanine were markedly increased in JD19 and LH3 under 100 mM NaCl treatment (Figure 6A,B). The content of tryptophan showed a trend of increase in LD2 (Figure 6C). Additionally, the levels of proline, citric acid, and threonine increased significantly in all cultivars.

To further confirm the key metabolic pathways in three groups under salt treatment, we carried out an enrichment analysis of DEMs. The bubble plot results showed that there were 23, 27, and 32 metabolic pathways in JD19, LH3, and LD2, respectively (Figure 6D–F). In terms of *p*-value and impact, alanine, aspartate, glutamate metabolism, beta-alanine metabolism, and TCA cycle were the top three metabolic pathways in JD19 and LH3 (Figure 6D,E). TCA cycle, glycine, serine and threonine metabolism, glyoxylic acid, and dicarboxylic acid metabolism were key metabolic pathways in LD2 (Figure 6F). Together, these results suggested that amino acid metabolism and the TCA cycle were crucial pathways in response to salinity stress.

### 2.7. Effects of Salt Stress on Amino Acid Contents and Nitrogen Metabolism for JD19, LH3, and LD2

To further verify the relationship between amino acid metabolism and salt stress, we examined free amino acid contents in all cultivars after salt treatment. The results showed that salt treatment induced a significant increase in the content of alanine, glutamine, proline, asparagine, and serine in JD19 and LH3. The content of tryptophan, tyrosine, and glutamate showed a similar increasing trend under the salinity stress condition in all three cultivars, while the content of lysine, isoleucine, and valine was decreased to different degrees (Figure 7A).

Since amino acids are an important form of nitrogen (N) in plants, we speculated that salt stress might affect N metabolism in three soybean cultivars. Plants obtain N mainly in the forms of nitrate (NO_3_^−^) and ammonium (NH_4_^+^) from the soil. After NO_3_^−^ is transported into the cytoplasm by its transporters (NRT1 and NRT2), it is sequentially reduced to NH_4_^+^ by nitrate reductase (NR) and nitrite reductase (NiR) [28,29]. However, the accumulation of NH_4_^+^ is toxic to plant cells [30]. Thus, NH_4_^+^ must be rapidly assimilated into amino acids by glutamine synthetase (GS) and glutamate synthase (GOGAT) [28,29,31,32]. We first analyzed the changes of NO_3_^−^ and NH_4_^+^ contents in three soybean cultivars under salt stress. A significant decrease in NO_3_^−^ and an increase in NH_4_^+^ were found in JD19, LH3, and LD2 under salt stress. However, the changes were different in the three cultivars. The NO_3_^−^ content in JD19 and LH3 was 13.5% and 14.6%, respectively, higher than that in LD2, while the NH_4_^+^ content in JD19 and LH3 was 15.4% and 16.0%, respectively, lower than that in LD2 (Figure 7B). Then, we determined the activities of nitrate reduction-related enzymes. Results showed that the NR activity exhibited little difference in JD19, LH3, and LD2 under the control condition. Although salt stress apparently inhibited the NR activity in three soybean cultivars, the effects of salt stress on NR activities in JD19 and LH3 were less severe than that in LD2 (Figure 7B). Similarly, NaCl treatment showed an inhibitory effect on NiR activity, which was decreased by 12.5%, 14.1%, and 38.9% in JD19, LH3, and LD2, respectively (Figure 7B). Moreover, we also examined the activities of ammonia assimilation-related enzymes GS and GOGAT. Results showed that NaCl treatment significantly decreased the GS activity in three soybean cultivars. Relative to JD19 and LH3, the GS activity in LD2 roots decreased by 7.4% (Figure 7B). In addition, the GOGAT activity was also decreased by NaCl treatment, but there was no obvious difference among the three cultivars (Figure 7B).

Salt stress also affects the total N content in many plants [33,34]. Thus, we examined the N content in three soybean cultivars. Under the control condition, there was no significant difference among JD19, LH3, and LD2. However, 100 mM NaCl enhanced the N content in JD19 and LH3 but not in LD2. The N content in JD19 and LH3 under 100 mM NaCl treatment was increased by 7.2% and 6.8%, respectively (Figure 8A).

NUE has two components, N utilization efficiency (NUtE) and N uptake efficiency (NUpE). Plant NUE is affected by many physiological processes (N uptake, metabolism, assimilation, and allocation) and soil nutrient factors, such as soil N availability [28]. In order to explore whether JD19 and LH3 are more effective than LD2 in N uptake and utilization under salt stress, both NUpE and NUtE were analyzed. As shown in Figure 8B,C, JD19 and LH3 outperformed LD2 with respect to NUtE and NUpE under 100 mM NaCl treatment. Specifically, NUtE was decreased by 47.8%, 48.3%, and 70.0% in JD19, LH3, and LD2, respectively (Figure 8B). Similarly, NUpE was decreased by 13.3%, 14.8%, and 31.9% in JD19, LH3, and LD2, respectively (Figure 8C). However, under the control condition, there was no significant difference in NUtE and NUpE among the three cultivars.

### 2.8. Effects of Salt Stress on Organic Acid Contents and TCA Cycle for JD19, LH3, and LD2

The results of metabolome revealed that the TCA cycle was another important metabolic pathway of soybean in response to salt stress (Figure 6D–F). In order to further verify the relationship between the TCA cycle and salt tolerance of soybean, the activities of key enzymes involved in the TCA cycle and the contents of organic acid were determined. As shown in Figure 9, the activities of pyruvate dehydrogenase (PDH), citrate synthase (CS), aconitase (ACO), isocitrate dehydrogenase (IDH), α-ketoglutarate dehydrogenase (α-KGDH), succinate dehydrogenase (SDH), and fumarase (Fum) were significantly reduced, but the malate dehydrogenase (MDH) activity was not influenced by salt stress (Figure 9B). In addition, the contents of α-ketoglutarate, malate, fumarate, isocitrate, and succinate were decreased, while the content of citrate was increased under 100 mM NaCl treatment. Interestingly, the content of oxalacetate decreased in JD19 but increased in LH3 and LD2 (Figure 9A).

The TCA cycle is an important pathway of plant aerobic respiration. The intermediate products produced in the reaction process provide carbon (C) skeleton, reducing power and ATP for plant growth and development. Under the salt stress condition, the contents of NAD^+^ and NADH in three cultivars decreased in varying degrees. In LD2, the contents of NAD^+^ and NADH obviously decreased by 13.0% and 29.7%, respectively, resulting in the decrease in NADH/NAD^+^ ratio by 19.1% under salt stress. Meanwhile, the content of NAD^+^ in JD19 and LH3 decreased by 21.6% and 21.2%, respectively. The content of NADH decreased by 15.0% and 14.8%, which led to a small 8.1% increase in the NADH/NAD^+^ ratio of JD19 and LH3 (Figure 10A–C). Combining the enzymatic analysis, the activity of the TCA cycle in JD19 and LH3 was higher than that of LD2 under salt stress. Moreover, NADH produced during the TCA cycle will be converted into ATP through the oxidative phosphorylation process for plant life activities. As shown in Figure 10D–F, salinity caused a significant decrease in ATP content, especially in LD2, but had little effect on ADP content in three cultivars, which led to a lower ATP/ADP ratio in LD2. Therefore, salt stress had a greater effect on ATP content and ATP/ADP ratio in LD2 compared with JD19 and LH3.

## 3. Discussion

Salt stress affects plant growth and development, eventually causes a severe decrease in crop yield. Soybean is the main source of plant protein and lipid for humans. However, soybean growth is seriously affected by salt stress, especially in seedlings [35]. Thus, studying the responses of soybean seedlings to salt stress may provide significant insights into the mechanisms underlying soybean salt tolerance. In this study, three soybean cultivars, JD19, LH3, and LD2, were used for physiological, transcriptomic, and metabolic analyses to reveal the mechanism of short-term salt stress adaptability in soybean.

In our study, the greater primary root length, the more lateral roots, and the more biomass were observed in JD19 and LH3 relative to LD2 after 4 d of 100 mM NaCl treatment (Figure 1). These observations suggested that JD19 and LH3 may have an excellent adaptation to the saline condition than LD2. Because salt stress can rapidly produce a large number of ROS to damage plant cells, reducing the accumulation of ROS is one of the adaptive mechanisms for plants to improve salt stress resistance [17,18]. The activities of antioxidant enzymes (SOD, CAT, and APX) increased significantly in JD19 and LH3 under salt stress. In contrast, the contents of MDA and H_2_O_2_ increased by a smaller degree than those in LD2 (Appendix A). Similarly, RNA-seq results indicated that *GmCAT5* is obviously up-regulated by salt stress in both JD19 and LH3 but not in LD2 (Figure 4A). Glutathione reductase (GR) are small molecule oxidoreductases that are involved in the process of defense against oxidative stress by directly scavenging ROS or redox regulating target proteins [36]. Under salt stress conditions, genes encoding GR were up-regulated in JD19 and LH3, indicating that they play a positive role in reducing cell oxidative damage caused by salt stress (Figure 4A). Similar studies have been reported in sesame, tomato, and Arabidopsis [37,38,39]. Moreover, some of the up-regulated DEGs were located in the pathway associated with glutathione metabolism. Glutathione S-transferase (GST), which participated in glutathione metabolism, is imperative in controlling redox balance and improving abiotic stress tolerance in plants [40]. In the current study, more *GSTs* were observed up-regulated in JD19 and LH3 under the saline condition, suggesting that the expression of *GSTs* was closely related to the salt resistance of soybean (Figure 4A).

Another major strategy for plants to enhance salt tolerance is to maintain ion homeostasis [18]. Many overlapped DEGs in JD19 and LH3 were involved in ion transmembrane transport processes, including organic cation transport, inorganic cation transmembrane transport, and transmembrane transport (Figure 3D). These processes contributed to reducing the intracellular ion accumulation caused by salinity and improving the stress tolerance of plants. Similarly, physiological data indicated JD19 and LH3 have lower contents of Na^+^ and Cl^−^, but higher content of K^+^ than LD2 under salt stress (Figure 2). Therefore, JD19 and LH3 had a higher capacity of antioxidant and ion homeostasis maintenance compared with LD2 when they underwent salt stress conditions.

Salt stress also causes disorder of physiological processes such as photosynthesis, respiration, hormone action, and nutrition metabolism [18,41,42]. Many studies demonstrated that plant salt tolerance is closely related to various metabolic processes during plant growth and development [43,44,45]. In our study, JD19, LH3, and LD2 samples were separated clearly under the control and salt conditions based on PCA and OPLS-DA models (Figure 5). These results indicated that salt stress was the main reason for the difference in metabolite clustering. Interestingly, different metabolite enrichment results confirmed that salt stress-activated alanine, aspartate, glutamate metabolism, and beta-alanine metabolism are the top two metabolic pathways in JD19 and LH3 (Figure 6D–F). Similarly, genes commonly up-regulated by salt in JD19 and LH3 were also mapped to multiple amino acid metabolism pathways (Figure 4B). The above results suggested that amino acid metabolism played an important role in soybean response to salt stress.

Previous studies have been reported that amino acid metabolism is inextricably bound with abiotic stress tolerance [22,23]. Proline serves as an osmotic regulator involved in plant response to abiotic stresses. It also induces the expression of salt stress-responsive proteins to enhance the salt tolerance of *Pancratium maritimum* L [46]. In our study, the proline content was significantly higher in JD19 and LH3 than in LD2 under salt treatment. Similarly, the increase in alanine, asparagine, glutamine, and serine alleviated salt stress through osmotic adjustment in JD19 and LH3 (Figure 7A). Moreover, some amino acids, such as leucine, isoleucine, lysine, valine, and threonine, produce energy through catabolism to cope with environmental stresses [47]. Lysine, valine, leucine, and isoleucine degradation pathways have been identified as the key factors for drought tolerance in Arabidopsis [48]. Our study indicated that the contents of lysine, valine, and isoleucine were also decreased more in JD19 and LH3 than in LD2. The content of leucine showed a consistent decreasing trend in JD19 and LH3 but increased in LD2 (Figure 7A). Together, these results indicated that the greater ability of amino acid metabolism regulation in JD19 and LH3 might be correlated with osmotic adjustment and energy metabolism coping with salinity stress.

N, one of the essential elements for plants, is the key component of nucleic acids, proteins, chlorophylls, and plant hormones [49,50]. It is well known that amino acids are an important form of organic N in plants. Asparagine and glutamine are involved in the primary and re-assimilation of NH_4_^+^ [24]. Moreover, asparagine plays a major role in the long-distance transport of N and acts as an N storage pool in plants under adverse environments [51]. Therefore, changes in amino acids also reflect the changes in N metabolism in plants. On the one hand, plants could enhance the tolerance to salinity by increasing N acquisition [52]. Wang et al. [53] reported that the NO_3_^−^ content was significantly increased in rice leaves under salt stress. Endophyte *E. gansuensis* increases the NO_3_^−^ content in host plant *Achnatherum inebrians* compared with - E- plants (without endophyte) under NaCl stress [34]. In poplar and wheat, salt stress also increased the NH_4_^+^ uptake [54,55]. Our present study showed that salt stress induced a more significant decrease in NO_3_^−^ and increase in NH_4_^+^ in LD2 than in JD19 and LH3 (Figure 7B). On the other hand, plants could enhance the tolerance to salinity by regulating the activity of N assimilation-related enzymes [53,56,57,58]. Surabhi et al. [59] reported that the decrease in NR and NiR activities was more significant in salt-sensitive mulberry than in salt-tolerant mulberry. Betti et al. [60] showed that GS plays an important role in abiotic stresses by increasing amino acid catabolism and producing protective N compounds. In JD19 and LH3, the activity of NR, NiR, and GS was higher than that in LD2 under salt treatment (Figure 7B). Thus, we speculated that the adaptive regulation of N metabolism might be one of the key mechanisms of salt tolerance in JD19 and LH3.

Salt stress has profound effects on nutrient contents, leading to nutrient metabolic disorders in plants [61]. In wild barley (*Hordeum brevisubulatum*), the N content is increased in roots as the NaCl concentration increases [62]. In this study, we also observed that salt stress increases the N content in the roots of JD19 and LH3 (Figure 8A). This may be because plant cells produce more ROS-scavenging enzymes under salt stress, and N is an essential component of all enzymes [63,64]. Plant NUE is a complex trait including NUpE and NUtE [28]. NUpE represents the capacity of a plant to obtain N from soil, which is closely correlated to the uptake of NO_3_^−^ and the root structure. NUtE is more dependent on N assimilation and storage in plants [34]. As shown in this study, although salt stress affected the N uptake and assimilation to different degrees in the three soybean cultivars, the effect in JD19 and LH3 was less severe than that in LD2 (Figure 8B,C). Therefore, the high N content and NUE is one of the adaptive strategies in JD19 and LH3 for salt stress.

TCA cycle was another crucial metabolic pathway among three cultivars under salt stress (Figure 6D–F). In this study, the contents of most organic acid, including TCA cycle intermediates α-ketoglutarate, malate, fumarate, isocitrate, and succinate, showed downward trends under 100 mM NaCl treatment in three soybean cultivars (Figure 9A). Consistent with this, the activities of most enzymes involved in the catalytic process of the TCA cycle decreased significantly under salt stress (Figure 9B). The reduction in organic acid, especially TCA cycle intermediates, might be due to the decrease in TCA cycle activity and the increase in the demand for osmotic regulation compounds to deal with salt stress [65]. Similar results have also been reported in halophytes, such as *Thellungiella halophile* [65], *Suaeda salsa* [66], and *Chenopodium quinoa* [67]. In addition, the metabolic phenotype of organic acid reduction implied the decrease in C flow from glycolysis (EMP) to the TCA cycle, resulting in the decrease in TCA cycle activity and ultimately reducing the production of NADH and ATP [68]. We found that the content of NADH significantly decreased by salt stress, especially in LD2, which led to the lower ratio of NADH/NAD^+^ in LD2 (Figure 10A–C). In general, the NADH/NAD^+^ ratio was used to evaluate the strength of the TCA cycle [69]. Therefore, the higher ratio of NADH/NAD^+^ in JD19 and LH3 indicated the higher TCA cycle activity under salt stress, which promoted the regeneration of NAD^+^ and then entered the next cycle. Subsequently, NADH produced by EMP and TCA cycle will be further transformed into ATP through respiratory electron transfer and oxidative phosphorylation for plant life activities. Li et al. [70] found that salt stress could inhibit the TCA cycle and significantly reduce ATP production in cucumber seedlings. In our study, salt stress had less effect on ATP content in JD19 and LH3 compared with LD2 (Figure 10E). Previous studies have shown that the decrease in TCA cycle activity and reducing power might be related to the decrease in plant growth rate under salt stress [45,71]. Physiological results also showed that the inhibitory effect of 100 mM NaCl treatment on root growth of LD2 was significantly stronger than that of JD19 and LH3 (Figure 1). Therefore, the regulation of organic acid metabolism and the TCA cycle was closely related to growth inhibition, which might be an energy-saving strategy for JD19 and LH3 under the saline condition. In addition, oxaloacetate is an important TCA cycle intermediate product produced by the transamination of aspartate, which can enter the next cycle [72]. In LH3 and LD2, the level of oxaloacetate increased with the salinity concentration, accompanied by the decrease in aspartate content. The results indicated that the increase in oxaloacetate level caused by salt stress might be due to the enhanced transamination of aspartate.

## 4. Materials and Methods

### 4.1. Plant Materials, Growth Conditions, and NaCl Treatments

The three soybean cultivars, JINDOU 19 (JD19), LONGHUANG 3 (LH3), LONGDOU 2 (LD2), have similar growth periods (Appendix A). JD19 and LH3 have large seeds and lodging resistance. LD2 was bred in 2005, with 4–5 branches and small seeds. Seeds of these cultivars were provided by Professor Guohong Zhang at Gansu Academy of Agricultural Sciences, Lanzhou, Gansu, China. The seeds were sterilized with 1% NaClO for 5 min and 2% H_2_O_2_ for 20 min, respectively. Then, they were washed with sterile water and germinated on an aseptic sponge and gauze for 48 h at 28 °C in the dark. The germinated seeds were transferred to 300 mL hydroponic pots and grown in 1/4-strength Hoagland liquid medium (pH 6.0) for 2 d. The soybean seedlings with 3 cm roots were selected and transplanted to 1/4-strength Hoagland solution with or without 100 mM NaCl for 4 d. The solution was replaced every day. The plants were cultured in a 25 °C greenhouse with a light/dark period of 16/8 h under 300 µmol m^−2^ s^−1^ photon flux density. After treatment for 4 d, roots were immediately used for analysis. The seedling roots were photographed with a Nikon digital camera, and the primary root length and lateral root numbers were analyzed using the Image J software. More than 15 seedlings per cultivar were analyzed (*n* = 15).

### 4.2. Determination of MDA and H_2_O_2_ Content and the Activities of Antioxidant Enzymes

The MDA content was determined according to the method of Hodges et al. [73]. A total of 0.5 g of soybean roots were extracted in 10% TCA (trichloroacetic acid) at 4 °C. After centrifugation for 10 min at 4000× *g*, the supernatant was incubated for 1 h with the same amount of 0.5% TCA at 95 °C. Then the absorbance was measured at 440, 535, and 600 nm. The H_2_O_2_ content was measured according to the method of Yang et al. [74]. A total of 0.5 g of root tissues were extracted with 2 mL of cooled 0.1% TCA. After centrifugation for 20 min at 12,000× *g* at 4 °C, the supernatant was incubated with 1 mL of 1 M KI and 0.5 mL of 0.1 M Tris-HCl (pH 7.6) for 1 h in the dark. Then the absorbance was immediately measured at 390 nm. The activity of SOD, POD, CAT, and APX was determined as previously described [74]. In each independent assay, 0.5 g of samples were used in each replicate.

### 4.3. Total RNA Extraction and qRT-PCR Analysis

The total RNA was extracted from soybean roots using a Plant RNA Extraction Kit (Thermo Fisher). The cDNA was synthesized from total RNA (1 µg) using the PrimeScript RT Reagent Kit (Thermo Fisher, Waltham, MA, USA), and qRT-PCR was performed using the SYBR PrimeScript RT-PCR Kit (Thermo Fisher, Waltham, MA, USA). The *GmACTIN2* was used as the internal control. Fold changes (2^−ΔΔCt^) were expressed relative to the control. The primer sequences used in this study are listed in Appendix A.

### 4.4. Transcript Profiling

Root tissues were sampled from the three soybean cultivars subjected to control and 100 mM NaCl treatment mentioned above and were frozen rapidly with liquid nitrogen. There were three replicates per group for RNA-seq experiments. The RNA extraction, cDNA library construction, sequencing, and quality control were carried out according to the method of Yang et al. [75]. The sequence alignment was based on the NCBI database of soybean (https://www.ncbi.nlm.nih.gov/genome/term=txid3847[orgn] (accessed on 5 November 2021)) using Hisat2 tools soft [76]. Gene expression was calculated by the FPKM (Fragments per kilobase of transcript per million fragments mapped) algorithm [77]. The differentially expressed genes (DEGs) were screened with FDR (false discovery rate) < 0.05 and fold change ≥2. Enrichment analyses of Gene Ontology (GO) based on the DEGs were performed by using the ClueGO plugin in Cytoscape [78]. KEGG (Kyoto encyclopedia of genes and genomes) pathway and GO enrichment analyses were based on *p*-value < 0.05.

### 4.5. Metabolite Extraction and GC-TOF-MS Analysis

About 0.01 g of frozen root tissues were used for metabolite extraction. Samples were extracted in 450 μL of pre-cooled methanol containing 10 μL of adonitol (0.5 mg/mL) as an internal standard. After vortex mixing for 30 s, samples were ground with a ball mill (35 Hz) for 4 min, followed by ultrasound exposure for 5 min in an icy water bath, and centrifuged at 4 °C for 15 min at 12,000 rpm. After dried under vacuum, the extracted samples were incubated with 60 μL of methoxyamination hydrochloride (20 mg/mL in pyridine) for 30 min at 80 °C, and then derivatized with 80 μL *N*, *O*-Bis (trimethylsilyl) trifluoroacetamide (BSTFA) for 1.5 h at 70 °C. Subsequently, metabolite compounds were analyzed using gas chromatography (Agilent 7890, Santa Clara, CA, USA) coupled with a time-of-flight mass spectrometer (GC-TOF-MS, LECO Chroma TOF Pegasus HT, Silver Spring, MD, USA). Chroma TOF (V4.3x, LECO, Silver Spring, MD, USA) software and the LECO-Fiehn Rtx5 database were used for raw data analysis [79,80].

### 4.6. Determination of N and Ion Contents

The total N content was measured using flow injection analysis (FIAstar 5000 Analyzer, FOSS, Höganäs, Sweden) with 0.1 g of dry powdered samples. Additionally, N utilization efficiency (NUtE) and N uptake efficiency (NUpE) were calculated according to the method of Wang et al. [81]. Briefly, NUtE was calculated as the total plant dry weight divided by N concentration (g^2^ TDW mg^−1^ N). NUpE was calculated as the total N accumulation divided by root dry weight (mg N g^−1^ RDW). Dry samples (0.1 g) were used for the determination of ion contents. The Na^+^ and K^+^ contents were determined by the method of Chen et al. [33]. The Cl^−^ content was determined by the method of Li et al. [82].

### 4.7. Determination of Amino Acid and Organic Acid Contents

Amino acid contents were measured using the previously reported method [80]. Briefly, 0.1 g (dry weight) of soybean roots were extracted with 0.5 M HCl and vortexed at 8000 rpm for 20 min. After sonication in a water bath at 25 °C for 20 min, the samples were centrifuged at 20,000× *g* for 20 min. Finally, the supernatant was transferred to a liquid chromatographic sample bottle, added to ISTD, and then diluted to 1 mL with 80% acetonitrile aqueous solution (LC/MS). Hydrophilic interaction chromatography (HILIC) was used to separate amino acids (Agilent InfinityLab Poroshell 120 HILIC-Z). The concentrations of amino acids were measured by the comparison of peaks and retention times with corresponding standards (Sigma-Aldrich, Saint Louis, MO, USA). Organic acid contents were determined according to the method of Coelho et al. [83] with minor modifications. A total of 0.1 g (dry weight) soybean roots were extracted with 1.5 mL ultrapure water. After vortex mixing for 20 min, the samples were sonicated in a water bath at 25 °C for 50 min and then centrifuged at 6000× *g* for 6 min. A total of 1 mL of supernatant for 0.22 μM water phase filter membrane was put into the liquid chromatography injection bottle, and the contents of organic acid in the sample were determined by high-performance liquid chromatography (HPLC) (EMpower3 2998 AcQuITY Arc, Boston, MA, USA).

### 4.8. Determination of NO_3_^−^ and NH_4_^+^ Contents and Activities of N Metabolism-Related Enzymes

0.5 g of soybean roots were used for the determination of NO_3_^−^ and NH_4_^+^ contents and activities of N metabolism-related enzymes. The NO_3_^−^ and NH_4_^+^ contents were determined following the method of Wang et al. [81]. The activity of NR was determined according to the method of Du et al. [84]. Soybean roots were extracted in an enzyme extraction buffer (100 mM Hepes-KOH (pH 7.5), 7 mM cysteine, 3% PVPP and 1 mM EDTA). NiR was extracted in an enzyme extraction buffer (50 mM PBS buffer (pH 8.8), 1 mM EDTA, 3% (*w/v*) bovine serum albumin and 25 mM cystein). The NiR activity was determined spectrophotometrically following the procedure of Wang et al. [81]. GS was extracted with the enzyme extraction buffer (50 mM Tris-HCl (pH 8.0), 400 mM sucrose, 2 mM DTT, and 2 mM MgSO_4_), and the GS activity was determined according to the method of Wang et al. [81]. For determination of the activity of NADH-GOGAT, root tissues were extracted in an extraction buffer (100 mM Tris-HCl (pH 7.6), 1 mM EDTA, 1 mM MgCl_2_ 6H_2_O and 10 mM β-Mercaptoethanol), and then centrifuged at 12,000× *g* for 20 min at 4 °C. The supernatant was used for the determination of NADH-GOGAT. NADH-GOGAT activity was determined following the method of Singh et al. [85]. The reaction buffer (100 mM α-ketoglutaric acid, 25 mM Tris-HCl (pH 7.6) and 10 mM KCl) was preheated to 30 °C in a water bath and then sequentially mixed with 0.9 mL of reaction solution buffer, 0.1 mL of 3 mM NADH, 0.5 mL of crude enzyme extract and 0.4 mL of 20 mM glutamine. The absorbance change of the mixture at 340 nm was measured and recorded within 3 min.

### 4.9. Determination of NAD^+^, NADH, ADP, ATP Contents, and Enzyme Activities Related to TCA Cycle

0.5 g of soybean roots were used for the determination of NAD^+^, NADH, ADP, ATP contents, and enzyme activities related to the TCA cycle. The NAD^+^ and NADH contents were determined following the method of Wang et al. [86]. ADP and ATP contents were assayed using ELISA kits (Shanghai Jiwei Biological Technology Co., Ltd., Shanghai, China) according to the method as described by the manufacturer’s instructions.

PDH activity was determined according to the method of Millar et al. [87]. Root samples were extracted in an enzyme extraction buffer (80 mM tetrasodium pyrophosphate (pH 7.5), 0.3 M mannitol, 10 mM K_3_PO_4_, 25 mM cysteine, 2 mM EGTA, 1% PVP and 1% BSA). After centrifugation 2000× *g* for 5 min, the supernatant was obtained and continued centrifuged 15,000× *g* for 20 min, then resuspended the precipitation with 1 mL of resuspension buffer (10 mM TES-KOH (pH 7.5), 0.3 M mannitol, and 0.1% BSA). After centrifugation at 30,000× *g* for 45 min, the supernatant was the crude enzyme solution of PDH. To determinate the PDH activity, a 100 μL crude enzyme solution was added to the reaction buffer (75 mM Hepes-NaOH (pH 7.5), 10 mM NAD^+^, 10 mM MgCl_2_, 10 mM cysteine, 0.2 mM CoA, 2 mM thiamine pyrophosphate) and incubated at 25 °C for 15 min. After the 20 mM, sodium pyruvate was added, the changes of absorbance value at 340 nm within 3 min were recorded. The activities of CS, IDH, Fum, and MDH were determined according to the method of Jenner et al. [88]. α-KGDH was performed according to the method of Pekovich et al. [89]. Root samples were extracted in an enzyme extraction buffer containing 20 mM Tris-HCl (pH 7.5), 1 mM EDTA, 1 mM DTT, 0.2 mM PMSF, 0.01% Triton X-100 and 0.02% sodium deoxycholate. After centrifugation at 10,000× *g* for 10 min at 4 °C, the supernatant was used for the determination of α-KGDH. The α-KGDH activity was determined following the method of Pekovich et al. [89]. Briefly, a 400 μL crude enzyme solution was added to the reaction buffer (50 mM MOPS (pH 8.0), 10 mM NAD^+^, 2 mM MgCl_2_, 0.16 mM CoA, 1.2 mM CaCl_2_, 0.04 mM rotenone and 0.5% Triton X-100) and incubated at 25 °C for 15 min. After the 20 mM α-ketoglutarate was added, the changes of absorbance value at 340 nm within 3 min were recorded. SDH was determined following the method of Schirawski and Unden [90].

### 4.10. Statistics Analysis

For statistical analysis of metabolites, five JD19 biological samples under 100 mM NaCl treatment and six other biological samples, including JD19, LH3, and LD2 with or without 100 mM NaCl treatment, were used separately. PCA and OPLS-DA were performed using the SIMCA software (version 15.0.2). Differentially expressed metabolites were identified using Student’s *t-test* (*p* < 0.05) and VIP (VIP > 1) based on the weighted sum of the squares of the OPLS-DA. The metabolic pathways were determined by MetaboAnalyst (http://metaboanalyst.ca/ (accessed on 5 November 2021)) based on the significantly differential metabolites. Heat map, amino acid, and organic acid cluster analysis were generated by R (version 3.2.2). Physiological data analyses were performed with SPSS 17.0, and statistical analyses were conducted with one-way ANOVA followed by Duncan’s multiple range test (*p* < 0.05). Each experiment was repeated at least three times. The values were expressed as mean ± SE.

## 5. Conclusions

In summary, two soybean cultivars, JD19 and LH3, are more salt-tolerant than LD2 because of their higher antioxidant enzyme activities, less ROS accumulation, and higher capacities of K^+^ and Na^+^ homeostasis under salinity. Transcriptomic and metabolomic analyses indicated that JD19 and LH3 have strong adaptive strategies to salt stress. First, increased N acquisition and assimilation are beneficial for JD19 and LH3 to accumulate amino acids. Second, more amino acid accumulation contributes to osmotic regulation and N reserve in JD19 and LH3. Third, high N reserve and NUE is advantageous to produce more ROS-scavenging enzymes for salt tolerance. Fourth, high TCA cycle activity in JD19 and LH3 helps in the production of organic acid, NADH, and ATP to support their growth under salt stress. These results may provide theoretical guidance for breeding salt-tolerant soybean cultivars and for sustainable utilization of saline soils.

## Figures and Tables

**Figure 1 ijms-22-12848-f001:**
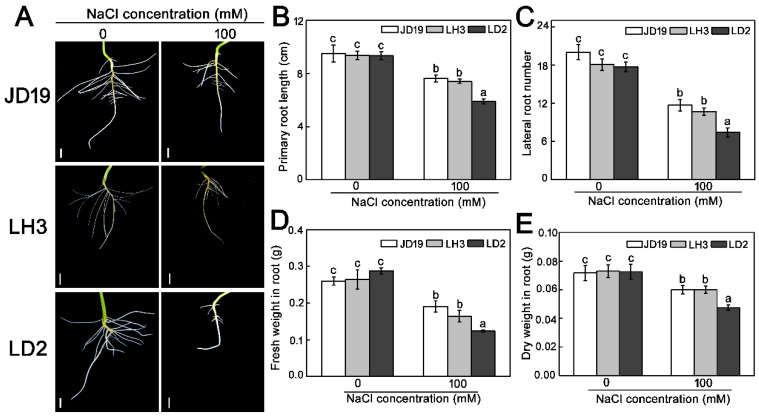
Root phenotype (**A**), primary root length (**B**), lateral root number (**C**), fresh weight (**D**), and dry weight (**E**) of three soybean cultivars under 100 mM NaCl treatment. Bar: 1 cm. Data are means ± SE (*n* = 3). Different letters indicate significant difference at *p* < 0.05.

**Figure 2 ijms-22-12848-f002:**
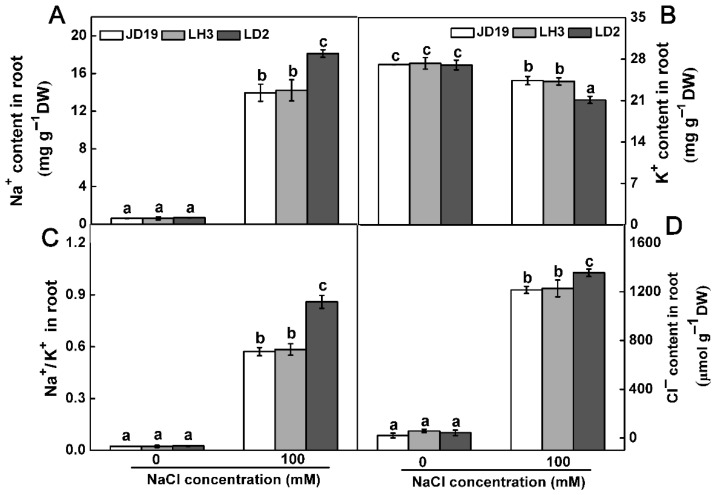
Na^+^ content (**A**), K^+^ content (**B**), Na^+^: K^+^ ratio (**C**), and Cl^-^ (**D**) content in roots of three soybean cultivars under 100 mM NaCl treatment. Data are the means ± SE (*n* = 3). Different letters indicate significant difference at *p* < 0.05.

**Figure 3 ijms-22-12848-f003:**
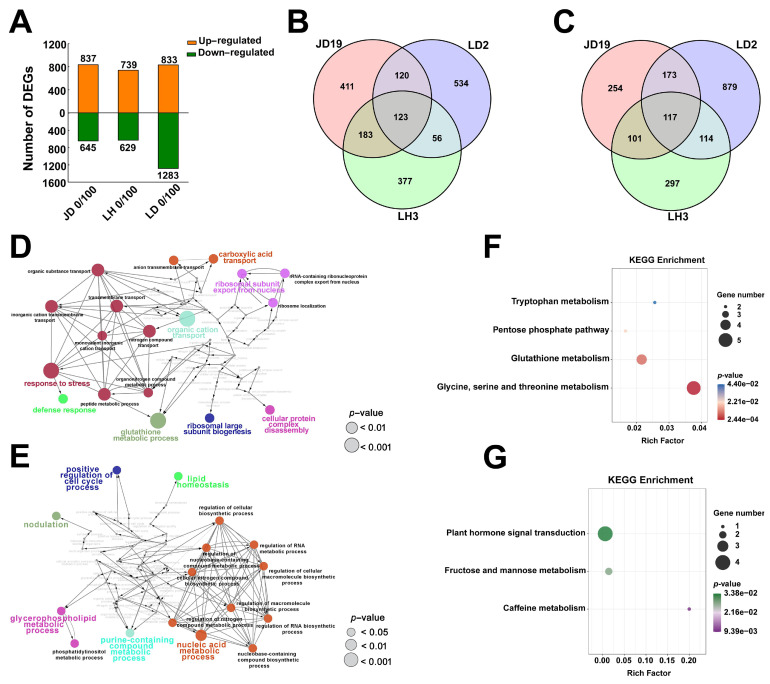
Transcriptome analysis of three soybean cultivars under 100 mM NaCl treatment. Statistics of DEGs (**A**), Venn diagram showing the overlapping and non-overlapping up-regulated DEGs (**B**), and down-regulated DEGs (**C**) among three soybean cultivars under 100 mM NaCl treatment. GO term of common up-regulated DEGs (**D**), and common down-regulated DEGs (**E**), KEGG pathway of commonly up-regulated DEGs (**F**), and common down-regulated DEGs (**G**) between JD19 and LH3 under 100 mM NaCl treatment. ClueGO analysis generated functionally grouped networks with GO terms as nodes linked based on their kappa score level (≥0.4). Colors reflect the label of the most significant term per group. The node size represents the term enrichment significance.

**Figure 4 ijms-22-12848-f004:**
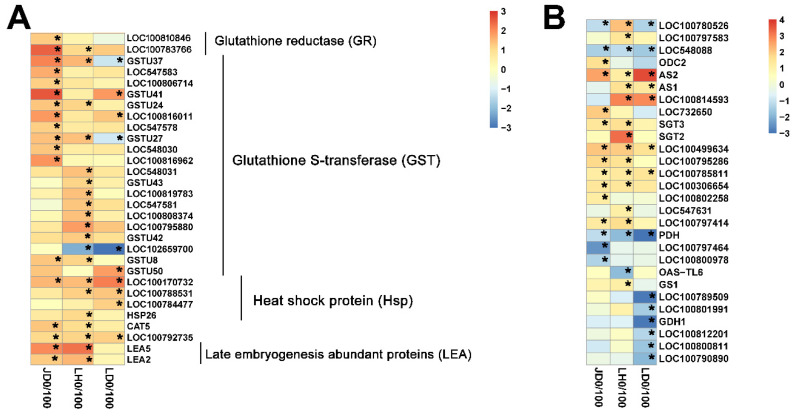
Heatmap of glutathione metabolism and stress response metabolism (**A**), amino acid metabolism (**B**)-related genes in three soybean cultivars under 100 mM NaCl treatment. Color panels display the Log2 fold change. Asterisks indicate the significant difference with an FDR < 0.01 and fold change ≥ 2.

**Figure 5 ijms-22-12848-f005:**
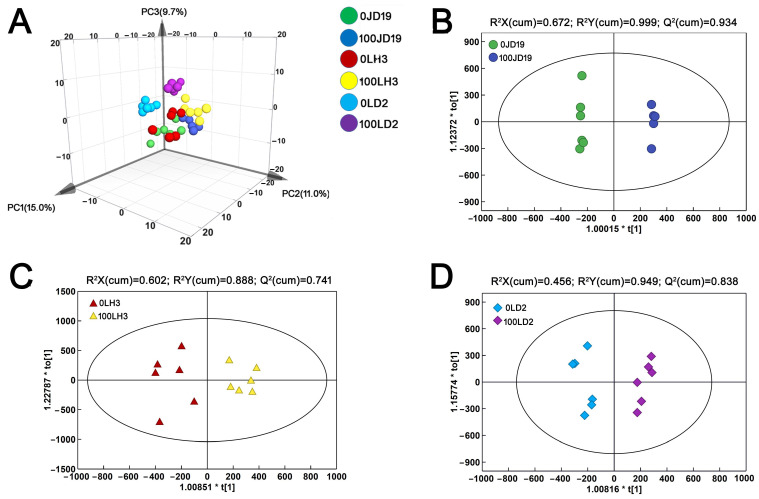
PCA diagram (**A**), and OPLS-DA scatter diagram of JD19 (**B**), LH3 (**C**), LD2 (**D**) under control and 100 mM NaCl treatment.

**Figure 6 ijms-22-12848-f006:**
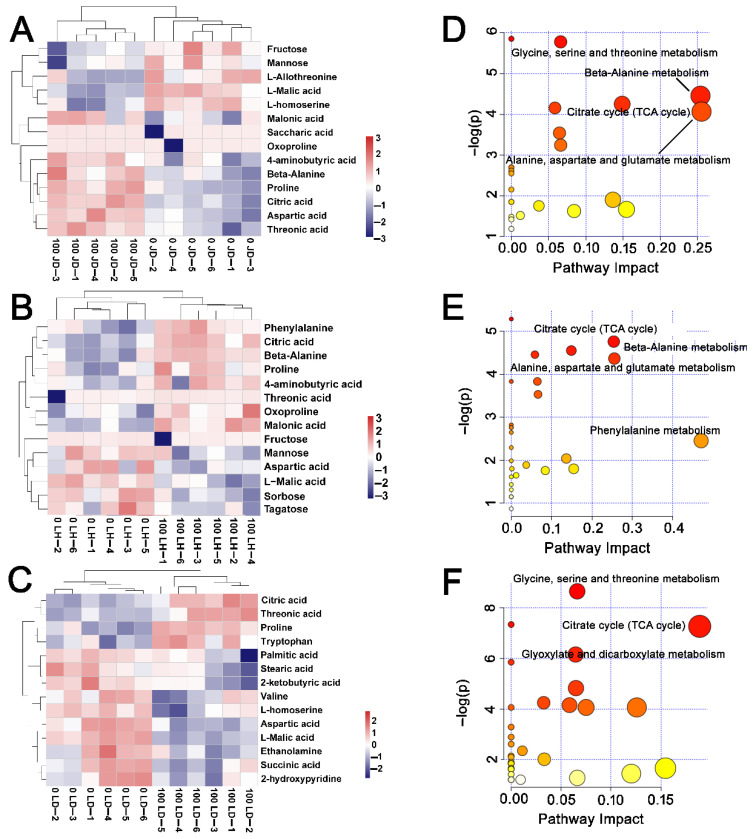
Heatmap clustering analysis of differentially expressed metabolites in (**A**) JD19, (**B**) LH3, (**C**) LD2, and KEGG pathway enrichment analysis in (**D**) JD19, (**E**) LH3, (**F**) LD2 under control and 100 mM NaCl treatment. Each bubble represents a metabolic pathway. The size of bubbles in abscissa represents the magnitude of the impact factor of the pathway in the topological analysis, with larger bubbles indicating higher impact factors. The bubble color in ordinate represents the *p* values of the enrichment analysis, with darker color indicating a more significant enrichment degree.

**Figure 7 ijms-22-12848-f007:**
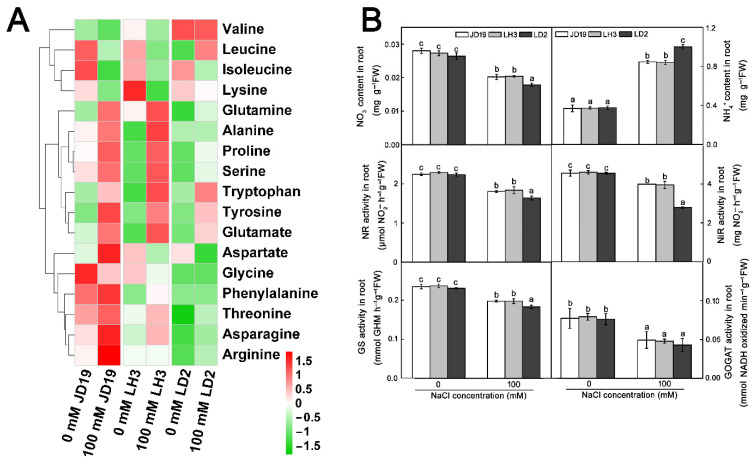
Effects of salt stress on free amino acid contents (**A**) and N metabolism-related enzymes activities (**B**) of three soybean cultivars. Data are means ± SE (*n* = 3). Different letters indicate significant difference at *p* < 0.05.

**Figure 8 ijms-22-12848-f008:**
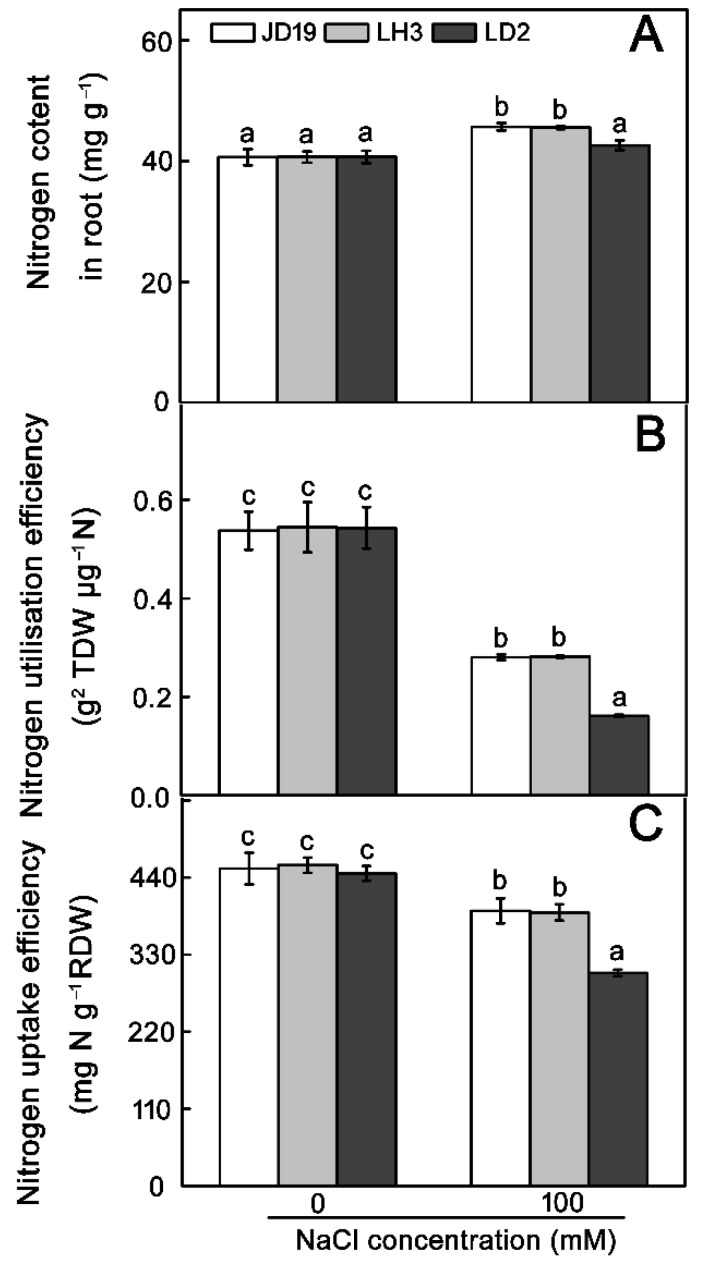
Effects of salt stress on N content (**A**), N utilization efficiency (**B**), and N uptake efficiency (**C**) of three soybean cultivars. TDW: total plant dry weight. RDW: root dry weight. Data are means ± SE (*n* = 3). Different letters indicate significant difference at *p* < 0.05.

**Figure 9 ijms-22-12848-f009:**
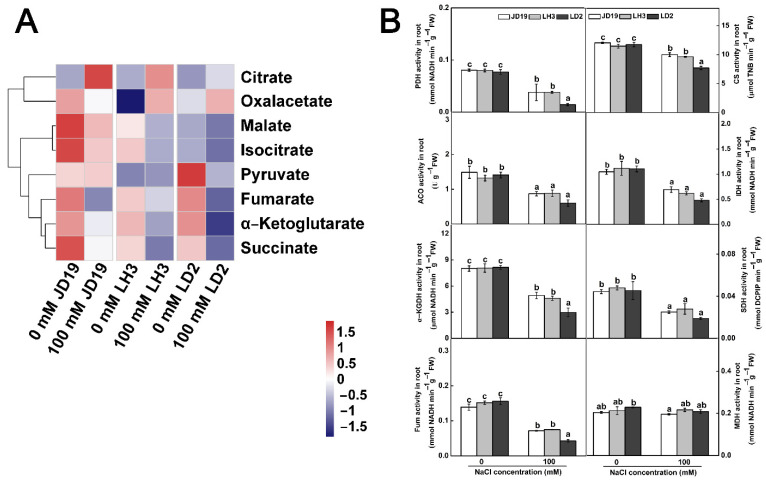
Effects of salt stress on organic acid contents (**A**) and TCA cycle-related enzymes activities (**B**) of three soybean cultivars. Data are means ± SE (*n* = 3). Different letters indicate significant difference at *p* < 0.05.

**Figure 10 ijms-22-12848-f010:**
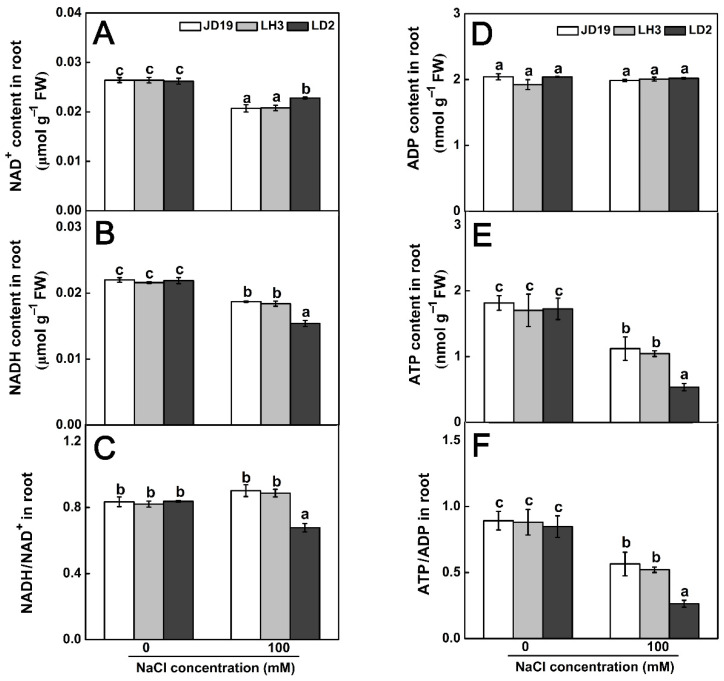
Effects of salt stress on NAD^+^ content (**A**), NADH content (**B**), NADH/NAD^+^ ratio (**C**), ADP content (**D**), ATP content (**E**), and ATP/ADP ratio (**F**) of three soybean cultivars. Data are means ± SE (*n* = 3). Different letters indicate significant difference at *p* < 0.05.

## Data Availability

The data presented in this study are available in the article and supplementary material.

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
