# Peer review of "Integrated Physiological, Transcriptomic, and Metabolomic Analyses Revealed Molecular Mechanism for Salt Resistance in Soybean Roots"

_ijms, 2021, doi:10.3390/ijms222312848_

Round 1

Reviewer 1 Report

In this manuscript, Jin and coworkers detail a study of the impacts of salt stress on three soybean cultivars which have different resistance to saline environments. This is an important area of investigation, as soybean is a primary source of plant protein and lipid to human diets, and plants which can tolerate saline conditions will become increasingly important as continued climate change models suggest and increasing land area with saline environments over the next century. Here, studies were performed at a number different levels: physiological, transcriptomic, metabolomic, and bioinformatics. Overall, the authors demonstrate convincingly that two of the cultivars, JD19 and LH3, are more saline tolerant than the other, LD2. The evidence gathered from a number of different measurements supports this conclusion, and includes lower levels of reactive oxygen species and higher levels of potassium and sodium ions during homeostasis. These two cultivars appear to adapt to saline conditions through increased N acquisition and assimilation into amino acids to control osmotic regulation and the nitrogen reserve, the latter of which is helpful in accumulating ROS-scavenging enzymes. In addition, these cultivars exhibit high TCA activity to produce organic acids, ATP, and NADH needed for plant growth. These results have implications for the breeding of saline-tolerant plants in general, and soybeans specifically, that is a critical need in 21st century agriculture. This reviewer is enthusiastic about the publication of this paper with minor revision. A few points and some typographical corrections are suggested below.

  1. The authors note that the saline stress comparison is made at one time point, 4 days after initiating growth in saline, vs. the control. The authors should make clear in the Discussion section that they have not performed a longitudinal study. Nonetheless, this does not detract from the importance of the results, though it would be interesting to note the effects from such a longitudinal study.
  2. Reference 79 cited by the authors is a review that details GC and LC methods coupled to mass spectrometry used in metabolomics studies on serum and plasma, and covers methods used by many authors. As such, the reference to this article in line 569 is not sufficient. More detail needs to be included on how the annotation of metabolites is made; presumably it is from the combination of retention time and the electron ionization fragmentation pattern compared to the database cited, but it is not clear from the current text how identification of the metabolites is made.

Small editorial points

  1. Line 46. “crop” should be “crops”
  2. Line 63. Insert “to” between “adapt” and “saline”
  3. Line 66. “amino acid” should be “amino acids”
  4. Line 80. Do not end the sentence with “tolerance”, simply change this to a comma and continue as the next sentence is otherwise a sentence fragment.
  5. Line 165. Insert “and” before “peptide”
  6. Lines 173-175. Instead of “While” use “Meanwhile” and insert “and” before “plant”
  7. Line 203. Insert “and” before “isoquinoline”
  8. Lines 204 and 206. “pathway” should be “pathways” in these two cases
  9. Line 220. Delete “which”
  10. Line 299. “amino acid” should be “amino acids”
  11. Line 512. “period” should be “periods”
  12. Line 537. Use “replicate” instead of “repeat”
  13. Line 586-587. “amino acid” should be “amino acids”
  14. Line 668. “helps to the production” should be “helps in the production”

Author Response

Dear Editor,

Thank you and the reviewers for the constructive comments for our article. We have revised the manuscript “Integrated physiological, transcriptomic and metabolomic analyses revealed molecular mechanism for salt resistance in soybean roots (ijms-1473606)” according to the reviewers’ comments. In the point-by-point responses attached below, reviewers’ comments are in black fonts and our responses are in blue fonts.

Reviewer #1:

  1. The authors note that the saline stress comparison is made at one time point, 4 days after initiating growth in saline, vs. the control. The authors should make clear in the Discussion section that they have not performed a longitudinal study. Nonetheless, this does not detract from the importance of the results, though it would be interesting to note the effects from such a longitudinal study.

>Answer:

Thanks for your comment. It has been supplemented in the Discussion section (L387-389).

  1. Reference 79 cited by the authors is a review that details GC and LC methods coupled to mass spectrometry used in metabolomics studies on serum and plasma, and covers methods used by many authors. As such, the reference to this article in line 569 is not sufficient. More detail needs to be included on how the annotation of metabolites is made; presumably it is from the combination of retention time and the electron ionization fragmentation pattern compared to the database cited, but it is not clear from the current text how identification of the metabolites is made.

>Answer:

Thanks for your comment. We have added the related references to explain more details (L570).

  1. Small editorial points

Line 46. “crop” should be “crops”

Line 63. Insert “to” between “adapt” and “saline”

Line 66. “amino acid” should be “amino acids”

Line 80. Do not end the sentence with “tolerance”, simply change this to a comma and continue as the next sentence is otherwise a sentence fragment.

Line 165. Insert “and” before “peptide”

Lines 173-175. Instead of “While” use “Meanwhile” and insert “and” before “plant”

Line 203. Insert “and” before “isoquinoline”

Lines 204 and 206. “pathway” should be “pathways” in these two cases

Line 220. Delete “which”

Line 299. “amino acid” should be “amino acids”

Line 512. “period” should be “periods”

Line 537. Use “replicate” instead of “repeat”

Line 586-587. “amino acid” should be “amino acids”

Line 668. “helps to the production” should be “helps in the production”

>Answer:

Thanks for your attention. We have corrected the grammatical error in the manuscript.

Reviewer 2 Report

Comments and Suggestions for Authors (18/11/21)

 1. Introduction:

Very good and clear

Good presentation and clarification on the research work

Enough for the introduction to work

2. Results:

Excellent presentation of results.

I have just only a question: why it you show the S-figures separately?

It becomes confusing for the reader..... Since they are mentioned so much in the text, I think it would be more convenient to put them in the text as well.

Regarding the tables, it is correct because they are all presented in the annex.

Point 2.3 is a little tricky to read - it is too long which distracts the reader, perhaps because the all figures are not in the text!

Also you repeat information presented in 2.2

3. Discussion:

References are very well introduced in the text.

And, does very well the connection of his work with the different authors

4. Materials and Methods

Good presentation and very clear

However, the material and methods chapter should be presented after the introduction of the work. One of the objectives of this chapter is to make the reader understand the results obtained.

Also in this chapter is where the author presents the corresponding acronyms that are not understood (in the results) without reading the Material and Methods first.

5. Conclusions:

Very clear and well written

References:

All references are in the text and are very diverse (years and authors).

The work is very strong and rich, as can be seen from the large number of references

GENERAL INFORMATION:

The paper is very, very good and very useful for the scientific community

I would advise to present a list with the acronyms

Author Response

Dear Editor,

Thank you and the reviewers for the constructive comments for our article. We have revised the manuscript “Integrated physiological, transcriptomic and metabolomic analyses revealed molecular mechanism for salt resistance in soybean roots (ijms-1473606)” according to the reviewers’ comments. In the point-by-point responses attached below, reviewers’ comments are in black fonts and our responses are in blue fonts.

Reviewer #2:

  1. Excellent presentation of results. I have just only a question: why it you show the S-figures separately?It becomes confusing for the reader..... Since they are mentioned so much in the text, I think it would be more convenient to put them in the text as well. Regarding the tables, it is correct because they are all presented in the annex.

>Answer:

Thank you for your advice. In the study, the data with low relevance to the theme was shown in the S-figures. On the one hand, the supplementary figures contribute to the better understanding of theme in the manuscript. On the other hand, the manuscript does not appear too long and messy.

  1. Point 2.3 is a little tricky to read - it is too long which distracts the reader, perhaps because the all figures are not in the text!Also you repeat information presented in 2.2

>Answer:

Thank you for your advice. In fact, point 2.2 were the transcriptional characteristics, included the numbers of DEGs, the venn diagrams and the correlation analysis between qRT-PCR and RNA-seq data in three soybean cultivars response to salt stress.

However, point 2.3 were the detailed explanation of venn diagrams (Fig. 3B-C) including the common up-regulated and down-regulated genes in all soybeans, the jointly up-regulated and down-regulated genes in two salt-tolerance genotypes, the independently up-regulated and down-regulated DEGs in three soybeans.

We used two databases, KEGG and GO, to dig much more information of DEGs. Although there might be a little long to read, it can offer more information to readers.

  1. However, the material and methods chapter should be presented after the introduction of the work. One of the objectives of this chapter is to make the reader understand the results obtained.Also in this chapter is where the author presents the corresponding acronyms that are not understood (in the results) without reading the Material and Methods first. I would advise to present a list with the acronyms

>Answer:

Thank you for your advice. In this journal, the material and methods chapter is behind the result and discussion chapters. Therefore we added a list with the acronyms in the manuscript (L39-42).
